# GENERATIVE SPATIAL REASONERS VIA REINFORCE-MENT LEARNING BASED INTRINSIC REFLECTION

## ABSTRACT

Recent advancements in image generation have achieved impressive results in producing high-quality images. However, existing image generation models still generally struggle with a "spatial reasoning dilemma", lacking the ability to accurately capture fine-grained spatial relationships from the prompt and correctly generate scenes with structural integrity. To mitigate this dilemma, we propose **RL-RIG**, a **R**einforcement **L**earning framework for **R**eflection-based **I**mage **G**eneration. Our architecture comprises four primary components, which follow a ***"Generate-Reflect-Edit"*** paradigm to spark the reasoning ability in image generation for addressing the dilemma. The process repeats the following steps: Generating a new image based on the input prompt and any previously generated image (if available), verifying whether the image satisfies all specified spatial relationships, and generating edit prompts when necessary. The training process is divided into two distinct stages: First, we employ Group Relative Policy Optimization (GRPO) to train the VLM Actor for edit prompts; second, we train the Image Editor for better image quality under a given edit prompt with GRPO. Unlike traditional approaches that solely produce visually stunning yet structurally unreasonable content, our evaluation metrics prioritize spatial accuracy, utilizing Scene Graph IoU and employing a VLM-as-a-Judge strategy to assess the spatial consistency of generated images on LAION-SG dataset. Experimental results exhibit that RL-RIG outperforms existing state-of-the-art open-source models by up to 11% in generating images with a controllable and precise spatial reasoning paradigm. [1]

## 1 INTRODUCTION

Recent years have witnessed remarkable progress in text-to-image generation such as Stable Diffusion 3.0 (AI, 2025), FLUX 1.0 (Labs, 2024), and Janus Pro (DeepSeek, 2025), which can produce high-fidelity and unprecedented images from textual prompts. However, they still face a "*spatial reasoning dilemma*", struggling to accurately capture and control fine-grained spatial relationships between objects within generated images while still producing visually impressive images. As illustrated in Figure 1, while Flux renders high-resolution pictures, it struggles to accurately represent the complicated spatial relationships specified in the prompt. This limitation becomes particularly critical in scenarios that require depicting complex spatial relationships rather than solely pursuing aesthetics (Chen et al., 2024a; Song et al., 2024).

As two pioneering works, ControlNet (Zhang et al., 2023) and GLIGEN (Li et al., 2023) have been proposed to guide the generation of outcome images. Despite these advancements, they still fall short in achieving flexible and precise control over relative spatial reasoning (Lukovnikov & Fischer, 2024; Zhao et al., 2023; Chen et al., 2024b). We attribute the reasons as follows: First, in addition to text descriptions, these methods often require additional user inputs, such as bounding boxes, keypoints, or reference images, which prevents them from realizing an end-to-end scenario. Second, as prompt complexity increases, these methods exhibit limited reasoning ability, making it difficult to accurately interpret all the spatial relationships described. For instance, the CLIP (Radford et al., 2021) encoder in most image generation models supports only at most 77 input tokens, and its training data primarily

---

[1]Code is available at `https://anonymous.4open.science/r/RL-RIG-demo-12AE/`

**ID:** 497747
**Prompt:** *stone tall tower attached to **stone arched bridge**; **wearing cloak standing person walking near wooden multi-story building**; red glowing lantern hanging from wooden multi-story building; **stone arched bridge leading to stone grand castle**; wooden sailing boat sailing on wooden sailing boat's reflection; Houses, graphics, Sailboats, **fantasy**, River, Castle.*

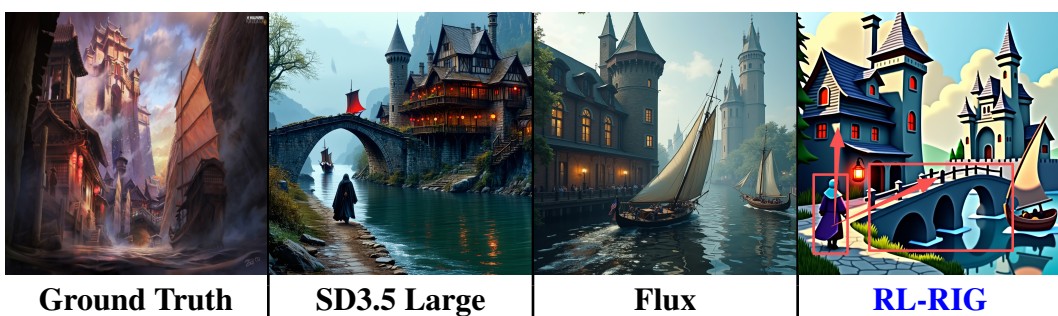

| **Ground Truth** | **SD3.5 Large** | **Flux** | **RL-RIG** |

**ID:** 523300
**Prompt:** *female old person walking towards **tall building**; **two-wheeled bike** leaning against tall building; wire basket attached to two-wheeled bike; **white fluffy dog sitting in wire basket**; male young person walking away from tall building; tall building have large sign; **female young person riding two-wheeled bike; female young person walking towards tall building**; Lee-Miller-in-paris-1944*

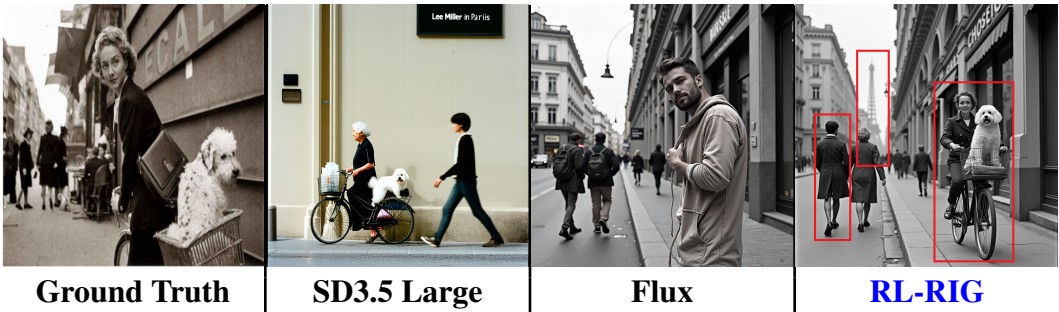

| **Ground Truth** | **SD3.5 Large** | **Flux** | **RL-RIG** |

**ID:** 511475
**Prompt:** ***tall illuminated lamp**; **tricolored flag on top of large illuminated building**; golden statue on top corner of large illuminated building; indistinct vehicle in front of large illuminated building; **indistinct person in front of large illuminated building**; Hanoi Hosts 2nd International Watercolour Painting Exhibition*

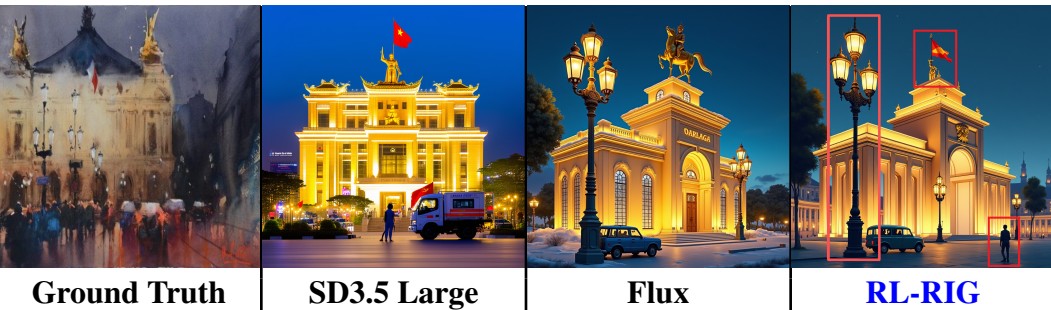

| **Ground Truth** | **SD3.5 Large** | **Flux** | **RL-RIG** |

Figure 1: Comparison of the generated image by Stable Diffusion 3.5 Large, Flux 1.0, RL-RIG **(choosing Flux as base model)**, and the ground truth image. Our method captures the relations in *blue* better, and compensates for the deficiencies in Flux's results. Actually, even the ground truth image does not comply to all the given relationships in the annotation.

focuses on word-level phrases (e.g., *a cat*, *a dog*) instead of global semantics and relational meanings (e.g., *"a small cute cat stands in front of a yellow dog wagging its tail"*).

Besides "spatial reasoning dilemma", evaluating spatial consistency in generated images is another challenging aspect. Traditional metrics, such as IS (Salimans et al., 2016), FID (Heusel et al., 2017), LPIPS (Zhang et al., 2018), CLIPS score (Hessel et al., 2021), and CMMD (Jayasumana et al., 2024), often focus on the distance between generated image and real images, while failing to assess spatial accuracy achieved by the generated image.

Addressing these limitations involves overcoming several key challenges. First, the complexity of spatial relationships makes it difficult for text encoders to interpret and represent them all at once. Second, existing image generators and editors, which focus little on spatial layout, tend to produce visually appealing results that lack fidelity to the original prompt. Third, the limited availability of dedicated datasets and evaluation methodologies for complex spatial relationships presents significant challenges for the training and development of frameworks targeting such objectives.

In this work, we propose **RL-RIG** (a **R**einforcement **L**earning framework for **R**eflection-based **I**mage **G**eneration), a novel **Generate-Reflect-Edit** framework that integrates reinforcement learning and reflection with vision-language models to enhance control in spatial relationships for text-to-image generation. Specifically, to leverage the actor-critic framework for spatial reasoning, we harness the strong reasoning capabilities of VLMs to serve as **an Actor and a Checker**. Based on them, we integrate **an Image Generator and an Image Editor**, as well as a reflection process for an image-refining loop. The training process involves two stages: training the VLM Actor using Group Relative Policy Optimization (GRPO (Guo et al., 2025)) with other components frozen, and training the Image Editor under the guidance of the VLM Checker via GRPO.

We reformulate text-to-image generation as sampling over **trajectories**, consisting of noise vectors, inversion states, edit actions, whose branching determines relational structure (Figure 2). Our RL-RIG activates the intrinsic reward ability for VLM when being as a judge, while Reinforcement Learning discourages low-advantage branches to increase the likelihood of scene-graph-consistent endpoints.

Our key contributions are:

- We identify the critical challenge of generating images with complex spatial relationships. Our analysis reveals that current baseline models either rely on supplementary inputs or exhibit constrained reasoning capabilities due to limitations in text encoders.

- To cope with the complexity of spatial relationships, we propose RL-RIG, a Self-Reflection based multi-agent framework to **Generate, Reflect and Edit** the generated image and trigger the **intrinsic reflection** ability of VLMs to **optimize the generation trajectory**, which enables the exhaustive resolution of each required spatial relationship.

- To address the limitation of current image editing frameworks, we explore test-time scaling and enabled **a Chain-of-Thought reasoning process for Image Generation**, as well as a two-phase RL featured by post-training the reflection Actor and the Image Editor.

- To address the lack of criteria for spatial relationships, we utilize Scene Graph IoU (Zhang et al., 2024) and VLM-as-a-judge (Lee et al., 2024) to **judge the image by faithfulness to the prompt**, instead of by the traditional "ground truth", showing an up to 11% increase in SG-IoU in the experiments.

## 2 RELATED WORKS

Recent work on spatially controllable image generation has focused on incorporating explicit layout or grounding inputs into diffusion models. Works like GLIGEN (Li et al., 2023), ControlNet (Zhang et al., 2023), layout-diffusion (Zheng et al., 2023) and FreestyleNet (Xue et al., 2023) have demonstrated effective ways to guide image generation based on specific spatial inputs. More recent approaches have extended these capabilities by adding style control (Wang et al., 2025c). Beyond bounding-box layouts, there are also scene-graph approaches (Farshad et al., 2023; Wang et al., 2025b). However, most of these techniques require supplementary information beyond text prompts, which limits their practical applicability in end-to-end scenarios.

Another thread of work applies Reinforcement Learning (RL) and preference optimization to steer diffusion models toward better alignment or specific objectives, such as DDPO (Black et al., 2023), Diffusion-DPO (Wallace et al., 2024), and Diffusion-RPO (Gu et al., 2024). These works illustrate that policy-gradient methods (PPO (Schulman et al., 2017), DPO (Rafailov et al., 2023), RPO (Yin et al., 2024), etc.) can be used to directly optimize diffusion models for complex rewards, analogously to RLHF (Ouyang et al., 2022) in vision language models (Pan & Liu, 2025). Nevertheless, few studies apply group reward-based preference optimization, like GRPO (Guo et al., 2025), into image generation models.

A related set of methods uses Vision Language Models or multi-step reasoning to improve control and alignment (Wen et al., 2023; Li et al., 2025a; Sun et al., 2025; Jiang et al., 2025). There has also been progress on rectified-flow generative models and diffusion inversion for more precise control and editing (Wang et al., 2024; Patel et al., 2024; Rout et al., 2024), although there are few approaches to leverage the reasoning ability of VLMs for better generative results.

Our work bridges these research directions by combining reinforcement learning with VLM-based reasoning in an intrinsic reflection framework that addresses the spatial reasoning challenges faced by current image generation systems, only requiring plain text input.

## 3 PRELIMINARIES

We focus on difficult text-to-image generation challenges where the input text is extremely sophisticated with many spatial relationships or other requirements. We formalize our problem as follows: let $Q = \{Q_1, Q_2, ..., Q_n\}$ be the set of all given text requirements, and $O = \{o_1, o_2, ..., o_m\}$ be the set of all objects with specific features. Each text requirement $Q_i$ can describe either a new object $o_i$ (with or without adjectives), a spatial relationship $(o_{x_i}, r_i, o_{y_i})$ between given objects, meaning that the object $o_{x_i}$ has a relationship $r_i$ over $o_{y_i}$, or a plain-text description $d_i$ for other properties of the image. Formally,

$$Q_i = \begin{cases} o_i, & Q_i \text{ is an object} \\ (o_{x_i}, r_i, o_{y_i}), & Q_i \text{ is a relationship} \\ d_i, & \text{otherwise} \end{cases} \tag{1}$$

The Image Generator $\mathcal{G}(Q) = f_{\theta,\xi}(x_T, T) = I$ takes in only the text prompts, and outputs an corresponding image subject to the requirements in the text. We suppose its output depends on the network parameter $\theta$ and the random seed $\xi$.

The Image Editor $\mathcal{E}(I, Q) = I'$ takes in an image and an edit prompt, and outputs the edited image. We suppose it is an inversion-based model, which contains inversion stage $f_{\theta,\xi'}^{-1}(x_t, t|\Phi)$ and reversion stage $f_{\theta,\xi'}(x_t', t|\Phi)$. Here, $t$ is the number of inversion steps, and $\Phi$ is the text embedding of $Q$.

To evaluate whether the requirements are all satisfied, we can employ a checker:

$$\mathcal{C}(I, Q) = \frac{|Q'|}{|Q|} \tag{2}$$

where $Q' \subseteq Q$ is the set of all satisfied requirements, and $|\cdot|$ means the number of requirements in the given set. In our definition, the checker outputs a score, which represents the proportion of satisfied requirements. Although the checker's output can influence the reward function, we do not utilize the concept of "Critic" as in Reinforcement Learning, because the Temporal Difference (TD) is hard to compute and propagate in this scenario.

Let $\tau = \{x_T, ..., x_0\} \cup \{a_1, ..., a_K\}$ denote a generation trajectory composed of latent states and edit actions. The Generator and Editor induce a stochastic transition $P(\tau)$. After determining $\tau$, $I$ is determined accordingly by the generation and editing process. Our objective is to maximize the expected scene-graph fidelity:

$$\max \sum_Q \mathbb{E}_{\tau \sim P}[\mathcal{C}(\tau, Q)]. \tag{3}$$

We employ other related components to rewrite this goal into a component-related form. Let $\mathcal{VLM}(I, Q) = Q'$ to be a Vision Language Model that takes in an image and text prompts, and output text responses. Here, we treat the set of requirements as plain text.

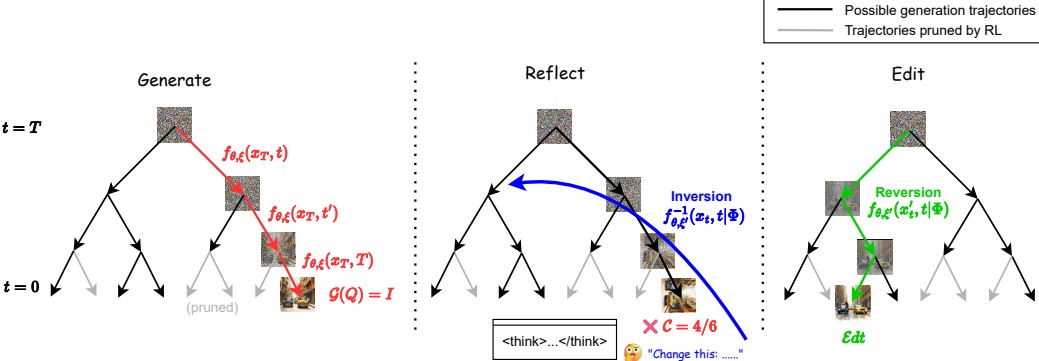

Figure 2: The Generate-Reflect-Edit framework, explained in a trajectory view. In each generation process, one of the possible trajectories is selected according to the random seeds. The VLM Checker will then reflect and check whether all parts of the prompt are satisfied. If not, the VLM Actor will provide an edit prompt and pass it to editor. Given the edit prompt, the Image Editor will perform inversion and reversion to explore a new possible trajectory. Dashed branches denote low-advantage trajectories pruned by GRPO. The 'Inversion' is actually performed in Edit stage.

We can then reformulate the objective of spatial image generation to be producing a multi-agent collaborative framework $\mathcal{F}_{\mathcal{G},\mathcal{E},\mathcal{C},\mathcal{VLM},\theta}(Q) = I$ with trainable parameters $\theta$ that satisfies as many requirements as possible:

$$\max \sum_Q \mathbb{E}[\mathcal{C}(\mathcal{F}_{\mathcal{G},\mathcal{E},\mathcal{C},\mathcal{VLM},\theta}(Q), Q)]. \tag{4}$$

# 4 METHOD

## 4.1 OVERVIEW

As shown in shown in Figure 2 and Figure 3, given a complex prompt with multiple spatial constraints, we devise RL-RIG, a **Generate-Reflect-Edit** framework that: **(1)** Generate an initial image, **(2)** Use a VLM Checker to identify unsatisfied spatial relationships, **(3)** Employ a VLM Actor to propose targeted edits, and **(4)** Apply these edits via an Image Editor. This process repeats until all constraints are satisfied or a maximum iteration limit is reached.

## 4.2 THE GENERATION PROCESS

The generation process of RL-RIG begins with an initial image $I = \mathcal{G}(Q)$ produced by a text-to-image Generator $\mathcal{G}$. The VLM Checker, based on a reasoning-finetuned VLM model, evaluates whether the generated image satisfies specified spatial relationships:

$$\Gamma_{Passed} = \begin{cases} True, & \text{if } \mathcal{C}(I, Q) = 1 \\ False, & \text{otherwise} \end{cases} \tag{5}$$

If the image fails this evaluation, the VLM Actor, trained from the VLM Checker, generates an editing prompt:

$$Q_{edit} = \mathcal{VLM}_{actor,\phi}(I, Q). \tag{6}$$

The Image Editor uses this prompt to modify the image:

$$I' = \mathcal{E}_\rho(I, Q_{edit}). \tag{7}$$

This process iterates through multiple cycles. In each cycle, the VLM Checker re-evaluates the edited image and the system applies further edits as necessary. The process continues until either the image satisfies all spatial criteria or the system reaches a predefined maximum number of editing steps.

## 4.3 THE TRAINING PROCESS

We observe that relying solely on pretrained modules is insufficient to generate images that fulfill complex spatial requirements. This limitation arises from two main issues: **(1)** The VLM Actor lacks knowledge of how to formulate effective edit prompts. **(2)** The Image Editor often struggles to modify images in ways that satisfy all specified relationships, regardless of the prompt provided. As a result, both components require post-training to improve their performance. We optimize them with GRPO to effectively compare candidate trajectory branches within a group and increase the mass on high-advantage branches.

Intuitively, we exploit GRPO as a trajectory-pruning mechanism to discourage unnecessary trajectories while preserving the ones that actually satisfy the given prompts. Group-wise relative advantages upweight edit prompts that raise scene-graph IoU upon re-sampling and downweight those that do not, which empirically shifts pass@k search gains into pass@1 improvements at similar compute (Yue et al., 2025).

To align with the definitions in Reinforcement Learning for GRPO, we can formulate the generation framework as a Markov Decision Process (MDP) defined by the tuple $(\mathcal{S}, \mathcal{A}, P, R)$, where the state $s_t \in \mathcal{S}$ encodes the current image $I$ and target scene graph, the action $a_t \in \mathcal{A}$ is either an initial generation prompt or an edit prompt, $P(s_{t+1} \mid s_t, a_t)$ is the (stochastic) image update via the generator or editor, $R(s_t, a_t) = r_t$ is the spatial-fidelity reward from the VLM Checker (Sutton & Barto, 2018).

As shown in Figure 4, the training process of RL-RIG contains two phases: **Training the VLM Actor only, and training the Image Editor only**. In both phases, the reward signal is provided by a fixed VLM Checker.

**Phase 1.** The first phase aims to address issue (1), training the VLM Actor for generating good edit prompts. First, we filter the images in the training set that fails the evaluation. Next, we let a group of $|G|$ VLM Actors generate the edit prompts separately using different random seeds:

$$a_t^i = \mathcal{VLM}_{actor,\phi,\xi^i}(s_t, a_0), \quad i \in [G] \tag{8}$$

where $i$ is the id of VLM Actor in the group, $a_t^i$ means the edit prompt $a_t$ generated by the $i$-th actor in the group (same meaning for the rest superscript), and $\xi^i$ is the random seed for each actor.

Then, we perform image editing based on each edit prompt, separately:

$$P(s_{t+1}^i|s_t, a_t^i) = \mathbb{E}_\epsilon[\mathcal{E}_\epsilon(s_t, a_t^i) = s_{t+1}^i], \quad i \in [G] \tag{9}$$

where $\epsilon$ is the random seed.

After that, we use VLM Checkers to evaluate the image one by one and get the reward score:

$$R(s_t^i, a_t^i) := r_t^i = \mathcal{C}(s_{t+1}^i, a_0), \quad i \in [G] \tag{10}$$

We want to maximize the expectation of the reward $r_t^i$. In this phase, only $\phi$ is trainable, so the goal is $\max \mathbb{E}_{i,\phi}[r_t^i]$. Following the approach of GRPO (Guo et al., 2025), we calculate the advantage and update the trainable parameters in VLM Actor using the same way:

$$\phi \leftarrow \phi + \alpha \nabla_\phi L^{\text{GRPO}}(\phi) \tag{11}$$

where

$$\mathcal{L}^{\text{GRPO}}(\phi) = \frac{1}{G} \sum_{i=1}^{G} \min\left(\rho_{t\_i} \widehat{A}_{t\_i}, \ \text{clip}(\rho_{t\_i}, 1-\epsilon, 1+\epsilon) \widehat{A}_{t\_i}\right) - \beta D_{\text{KL}}\left[\pi_\phi \mid \pi_{\text{ref}}\right]. \tag{12}$$

Here:

$$\mu_r = \frac{1}{G} \sum_{j=1}^{G} r_t^j, \quad \sigma_r = \sqrt{\frac{1}{G} \sum_{j=1}^{G} (r_t^j - \mu_r)^2}, \tag{13}$$

$$\widehat{A}_t^i = \frac{r_t^i - \mu_r}{\sigma_r}, \quad \rho_i = \frac{\pi_\phi(a_t^i \mid s_t^i)}{\pi_{\text{ref}}(a_t^i \mid s_t^i)}. \tag{14}$$

and $\pi$ is $\mathcal{VLM}$'s probability of taking such action.

**Phase 2.** In the second phase, we aim to address issue (2) by improving the performance of Image Editor after the VLM Actor performs relatively well. Therefore, we freeze the VLM Actor and train the DiT in the Image Editor. After filtering the image, we use one trained VLM Actor to generate the edit prompt, and employ a group of image editors to perform image editing. Then, we use VLM Checkers to evaluate each edited image in group one by one:

$$a_t = \mathcal{VLM}_{actor,\phi,\xi}(s_t, a_0), \tag{15}$$

$$P(s_{t+1}^i | s_t, a_t^i) = \mathbb{E}_\epsilon[\mathcal{E}_{\rho,\epsilon}(s_t, a_t^i) = s_{t+1}^i], \quad i \in [G] \tag{16}$$

$$r_t^i = \mathcal{C}(s_{t+1}^i, a_0), \quad i \in [G] \tag{17}$$

where $\xi$ and $\epsilon$ are random seeds, and $\rho$ is the trainable parameter for the Image Editor. The training goal of this phase is $\max \mathbb{E}_{i,\rho}[r_t^i]$. Calculating the advantage and updating the parameters $\rho$ in DiT is similar to the first phase, and omitted here.

## 5 EXPERIMENTS

### 5.1 EXPERIMENTAL SETUPS

**Base models.** In practice, we use Curr-ReFT (Deng et al., 2025b), a reasoning model from Qwen 2.5 as the base VLM model for a better reasoning ability. The Image Generator employs Flux (Labs, 2024), a state-of-the-art open-source image generation model, and the Image Editor employs RF-Inversion (Rout et al., 2025), a Flux-based model enabling efficient inversion and editing of real images based on given text descriptions without requiring a mask area. The GPRO training framework is modified from VLM-R1 (Shen et al., 2025; Face, 2025). In the future, all these components can be substituted by a stronger model if they are available for the same task.

**Environment Setup.** We set up a distributed training on $2 * 4$ A100 GPUs. We preprocess all the image generation process before the training begins. For training phase 1, we load 3 parallel VLM Actors and 1 VLM reference model on the first machine; we load 1 VLM Checker and 3 parallel image editors on the second machine. For training phase 2, we load 1 VLM Actor and 3 parallel image editors on the first machine; we load 1 VLM Checker, 1 Image Editor, 1 reference Image Editor and the trainer on the second machine. The training time for each phase is about 2 days. All the parallel Actors and Editors are the same networks, acting as load balancers.

**Hyper-parameters.** We set most the base models' hyper-parameters by default. To enable a stronger image-editing ability, we follow the advice in (Wang et al., 2024) and set $\gamma = \eta = 0.7$ and $Guidance\_scale = 10$ in rf-inversion. In the generation phase, each picture can be edited at most 10 times. If it still fails to pass the checker after 10 edits, a restart attempt will be made, which will restart the editing procedure from the original picture with a different random seed for all base models. If 3 restart attempt fails, the original generated picture will be saved. The optimal number of edit prompts is determined empirically: increasing the count from 10 to 11 yields no performance gain, while reducing it to 9 results in diminished performance. The same holds for the number of restart attempts.

**Dataset.** We utilize LAION-SG (Li et al., 2024) as the training and test dataset, a remarkably rare text-to-image dataset whose prompts feature **highly intricate spatial relationships**. Custom text-to-image datasets (like COCO (Lin et al., 2014) and DrawBench (Saharia et al., 2022)) are not used as they do not contain such complex spatial relationships in the prompt. The data split is set by default. For convenience, we select top 2000 images for training and top 500 images for testing by the aesthetic score. We use the same approach as described in LAION-SG to utilize GPT-4o for extracting the scene graph and evaluating the IoU.

**Evaluation metrics.** We observe that ground truth images do not align perfectly with the provided spatial descriptions, and thus applying traditional metrics like FID and IS would lead to biased results. Instead, we use metrics like Scene Graph IoU (Zhang et al., 2024) and VLM-as-a-judge (Lee et al., 2024) that can evaluate the images' loyalty to the extracted relationships. Suppose $Q$ is the Scene Graph extracted from ground truth (i.e., the text requirements), and $Q'$ is the Scene Graph extracted

Table 1: Comparison of different models on top-500 test subset of LAION-SG dataset. RL-RIG (raw) is an ablation study, meaning the framework of RL-RIG that does not go through two training phases. The highest score for each metric is **bolded**, and the second highest score is underlined.

| Method | SG-IoU | Ent-IoU | Rel-IoU | Qwen-Judge | GPT-Judge |
|---|---|---|---|---|---|
| SD3.5 Large | 0.2955 | 0.8145 | 0.7056 | 0.7896 | 0.6908 |
| Flux | 0.3319 | 0.9000 | 0.7714 | 0.7993 | 0.7256 |
| LAION-SG | 0.2618 | 0.7805 | 0.6858 | 0.6388 | 0.5133 |
| RL-RIG (raw) | 0.3575 | **0.9109** | 0.7770 | 0.7986 | 0.7212 |
| **RL-RIG** (post-trained) | **0.3699** | 0.9059 | **0.7837** | **0.8219** | **0.7322** |
| Ground Truth | 0.3864 | 0.9304 | 0.8129 | 0.7240 | 0.7536 |

Table 2: Performance improvements of RL-RIG (post-trained) over baselines on the LAION-SG dataset (top-500 test subset). Each entry shows absolute gain ($\Delta$) and percentage gain (%). Light green = improvement, light red = decline.

| Baseline | SG-IoU | | Ent-IoU | | Rel-IoU | | Qwen-Judge | | GPT-Judge | |
|---|---|---|---|---|---|---|---|---|---|---|
| | $\Delta$ | % | $\Delta$ | % | $\Delta$ | % | $\Delta$ | % | $\Delta$ | % |
| **SD3.5 Large** | +0.0744 | 25.2% | +0.0914 | 11.2% | +0.0781 | 11.1% | +0.0323 | 4.1% | +0.0414 | 6.0% |
| **Flux** | +0.0380 | 11.5% | +0.0059 | 0.7% | +0.0123 | 1.6% | +0.0226 | 2.8% | +0.0066 | 0.9% |
| **LAION-SG** | +0.1081 | 41.3% | +0.1254 | 16.1% | +0.0979 | 14.3% | +0.1831 | 28.7% | +0.2189 | 42.7% |
| **RL-RIG(raw)** | +0.0124 | 3.5% | -0.0050 | -0.6% | +0.0067 | 0.9% | +0.0233 | 2.9% | +0.0110 | 1.5% |

from the generated image by GPT-4o (OpenAI, 2024). Then, Scene Graph IoU is defined as the IoU between them:

$$IoU_{SG} = \frac{Q \cap Q'}{Q \cup Q'} \tag{18}$$

Suppose $O = \{o_i \in Q\}$ is all the objects that appear in $Q$, and $O' = \{o_i' \in Q'\}$ is all the objects that appear in $Q'$. Similarly, $Rel = \{r_i|(o_{x_i}, r_i, o_{y_i}) \in Q\}$ and $Rel' = \{r_i'|(o_{x_i}', r_i', o_{y_i}') \in Q'\}$ is all the relationships in generated image and ground truth image, separately. Then, the entity IoU and relationship IoU are defined as:

$$IoU_{Ent} = \frac{O \cap O'}{O \cup O'}, \quad IoU_{Rel} = \frac{Rel \cap Rel'}{Rel \cup Rel'} \tag{19}$$

**Unlike traditional metrics that measure image similarity, our IoU-based evaluation specifically targets spatial relationship fidelity.** This approach enables us to differentiate between images that merely appear visually similar to ground truth and those that accurately represent the specified spatial relationships. Traditional metrics like FID are inadequate for this task as they fail to capture the critical spatial configurations that define success in our problem setting.

The score of VLM-judge is defined in the same way as Equation 2. Here, we employ ReFT (a reasoning VLM based on Qwen2.5) and GPT-4o as two VLM judgers for meta-evaluation, separately.

## 5.2 RESULTS

We choose SD3.5 Large, Flux and LAION-SG as baseline models. The former two models are state-of-the-art open-source image generation models; the latter one is the model from the original literature of the LAION-SG, which is specially post-trained on the LAION-SG dataset. We compare the performance of RL-RIG before and after post-training to better understand its influence.

The result of our experiment is shown in Table 1 and Table 2. RL-RIG consistently outperforms all baselines across spatial reasoning metrics. The framework alone (raw) provides modest improvements (+3.5% SG-IoU over Flux), while post-training yields substantial gains (+11.5% SG-IoU). Notably, our method shows the largest improvements on the most challenging spatial relationships, validating our iterative refinement approach. We observe a decrease in Ent-IoU after post-training, which might be attributable to the limited ability of the Image Editor (**more success and failure cases can be found in Appendix C**).

### 5.3 MECHANISM: INTRINSIC REFLECTION AS TRAJECTORY-LEVEL REWARD

We observe that RL-RIG's *Generate-Reflect-Edit* loop converts internal evaluative signals into implicit rewards purely by VLM Checker's capability of judgment instead of external gold verifiers. We investigate the essential reason of this to be **Intrinsic Reflection**.

*Intrinsic Reflection* refers to the phenomenon where LLMs can learn to reason by only recursively optimizing self-generated feedback (Rafailov et al., 2024b; Wang et al., 2025a). This works because (Li et al., 2025b) suggests that reward modeling stage can be replaced by a principled method of eliciting the knowledge already captured during pre-training. In practice, RLIF (Zhao et al., 2025) and RLPR (Yu et al., 2025) replace external rewards with the model's own confidence or token-likelihood signals while still improving LLM's performance. This suggests that Large Language Models already embed a *generalist reward model* on thoughts and responses trajectories.

Our design **extends this idea to the multi-agent VLM setting**. The *VLM Checker* computes a scalar *intrinsic reward* $R(s_t^i, a_t^i)$ from the agreement between **(i)** the prompt-induced scene graph $a_0$ and **(ii)** the image's relations $s_{t+1}^i$. This reward then drives GRPO updates to the VLM Actor that proposes structured edit prompts, and subsequently to the Image Editor that executes geometry-preserving edits. In effect, reflection supplies a **dense, verifiable-by-the-model** signal for *spatial credit assignment*, which eventually pushes the policy to increase the probability of edits that raise scene-graph IoU and reduce relation violations.

Inspired by the idea in (Rafailov et al., 2024a), we can also view the model as a latent **Q-function** over action sequences (edit steps). Based on that, our goal is to couple it with GRPO's group optimization and optimize the whole system multimodally. The reflection step *measures* relational structure, the intrinsic reward *targets* it, and GRPO *amplifies* successful local edits into globally consistent layouts, which explains why RL-RIG particularly improves fine-grained spatial relations. This internal preference-based training can therefore deliver structurally faithful images rather than merely photorealistic ones.

## 6 LIMITATIONS AND FUTURE WORK

While our proposed RL-RIG framework demonstrates impressive capabilities, its performance is naturally influenced by the underlying base model, though all the post-training processes can alleviate this influence. Additionally, due to the pioneering nature of our work in addressing complex prompt reasoning, a comprehensive text-to-image dataset for such advanced and difficult spatial relationships is notably rare and remains in development.

Our research opens several promising avenues for future work. First, integrating more advanced reasoning mechanisms could further enhance the VLM Actor's ability to generate precise edit prompts. Second, developing specialized Image Editors that maintain better contextual consistency across multiple edits could address the discontinuity issues observed in Figure 6. Finally, creating a dedicated dataset for text-to-image tasks with extremely complex spatial relationships would greatly help this future work.

## 7 CONCLUSION

In this paper, we addressed the "spatial reasoning dilemma" in text-to-image generation by proposing RL-RIG, a Generate-Reflect-Edit framework that combines reinforcement learning with vision-language models to enhance spatial reasoning ability for complex scene generation. It frames spatial image generation as trajectory selection and shows that intrinsic reflection plus GRPO is a practical test-time scaling recipe to prune low-fidelity branches and realize scene-graph-faithful images. Experimental results on the LAION-SG dataset demonstrate that RL-RIG outperforms state-of-the-art models like Stable Diffusion 3.5 Large and Flux in terms of spatial coherence, as measured by metrics like Scene Graph IoU and VLM-based evaluation. RL-RIG is flexible in that all its components can be substituted by the latest state-of-the-art model.

We hope this work highlights the importance of reasoning in complex image generation tasks, and provides an insightful intrinsic reflection multi-agent framework approach to overcome the complex spatial relationships in such scenarios.

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

# A    ILLUSTRATION OF RL-RIG

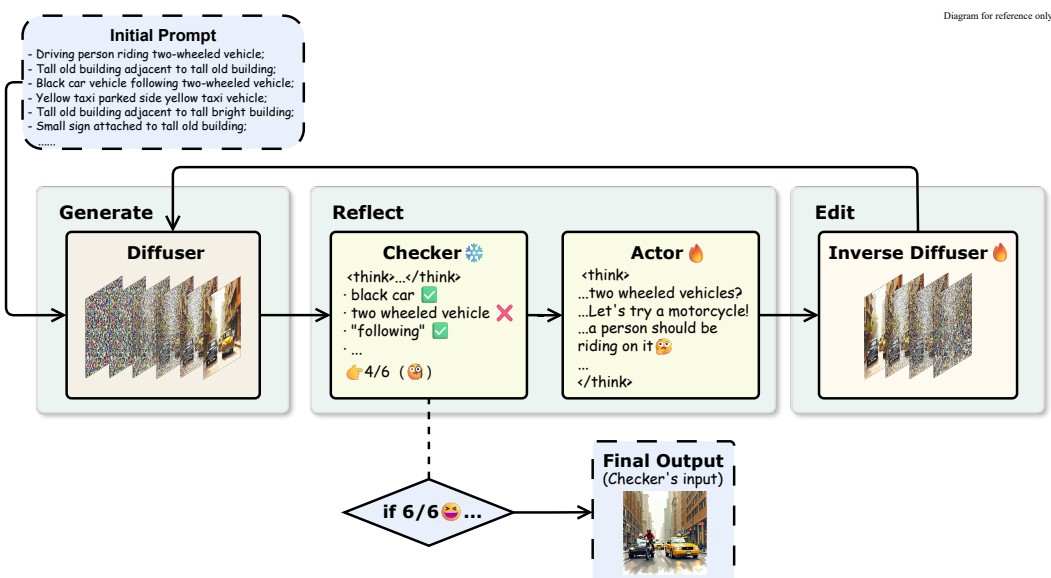

Figure 3: The overview of RL-RIG, which abides a Generate-Reflect-Edit paradigm and two training phases. Here, we suppose the Image Editor is composed of an inverse diffuser and a diffuser.

# B    DETAILS OF TWO-PHASE TRAINING

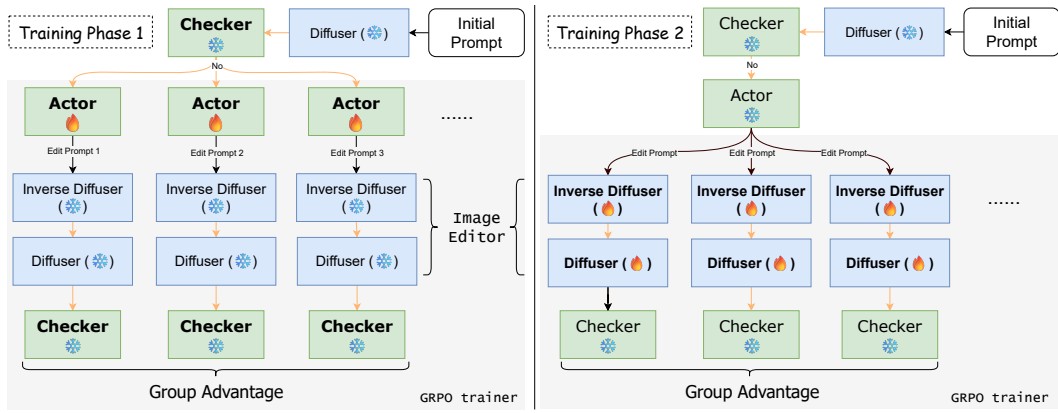

Figure 4: Illustration of two-phase training. For each phase a batch of responses are sampled, and group advantage is calculated by GRPO.

Our optimization only includes 2 phases, as is shown in Figure 4.

In the first training phase, the VLM Actor generates a batch of edit prompts. Each edit prompt is passed to a frozen Image Editor, and the corresponding image is graded by a VLM Checker. The checker's response is used as the reward function in GRPO to encourage it to generate an appropriate edit prompt.

In the second training phase, the VLM Actor is frozen, and a fixed edit prompt is passed to the Image Editor to generate a batch of image edits. The corresponding image is graded by a VLM Checker, whose response is used as the reward function in for refining the Image Editor's performance under the given prompt.

For the sake of generalization, we keep the VLM Checker untrained. This is because training it would bring about unnecessary bias to its feedback results. Once influenced by the training objective, the

reward system would become distorted and overfit to its bias. To further confirm the reliability of the base VLM model, we manually examined 50 cases and found that all of VLM checkers' outputs are aligned with human senses. Therefore, we conclude that the base VLM model shows a good zero-shot ability with regard to judging how many requirements are satisfied in the image, and that additional post-training on the VLM Checker would be potentially redundant.

The VLM Actor, on the other hand, does require post-training due to its functions in the system. We expect to yield edit prompts from the actor that could bring about images with exact spatial relationships, but we find that a tiny change in edit prompts would result in significantly different results. Notably, even a human-written edit prompt may not yield the desired result in many cases, sometimes performing worse than the post-trained VLM Actor, This shows the necessity of post-training the VLM Actor so as to cooperate well with the features of the Image Editor, its partner in the image generation process.

---

**Prompt:** *driving person riding two-wheeled vehicle; tall old building adjacent to tall old building; black car vehicle following two-wheeled vehicle; yellow taxi vehicle parkedside yellow taxi vehicle; tall old building adjacent to tall bright building; small sign attached to tall old building; A new day, Mumbai by Ramesh Jhawar, Impressionism Painting, Watercolor on Paper, Brown color*

---

| Agent | Response |
|---|---|
| **Image Generator** | Generated with the original prompt |
| **Checker** | \<think\> Let's analyze each description one by one: 
 1. *"black car vehicle following two-wheeled vehicle"*: There is no two-wheeled vehicle in the image, so this description is not satisfied. 
 2. *"yellow taxi vehicle parkedside yellow taxi vehicle"*: There are two yellow taxis parked next to each other, so this description is satisfied. 
 3. *"small sign attached to tall old building"*: There is a small sign visible on one of the tall buildings, so this description is satisfied. 
 4. *"tall old building adjacent to tall old building"*: There are two tall buildings next to each other, so this description is satisfied. 
 5. *"driving person riding two-wheeled vehicle"*: There are no two-wheeled vehicles in the image, so this description is not satisfied. 
 6. *"tall old building adjacent to tall bright building"*: There is a tall old building next to a taller bright building, so this description is satisfied. 
 \</think\>\<answer\> 4 \</answer\> |
| **Actor** | \<think\>...\</think\>\<answer\>*yellow taxi vehicle parked, small sign attached to tall old building, tall old building adjacent to tall old building, tall old building adjacent to tall bright building, black car vehicle following two-wheeled vehicle, watercolor*\</answer\> |
| **Image Editor** | Performed editing with the given prompt |
| **Checker** | \<think\>...\</think\>\<answer\> 6 \</answer\> |

Figure 5: A succeeded trial of image generation by RL-RIG, with the input prompt (id=523378) and the reasoning process. After reflection, the actor successfully guides the Image Editor to add a two-wheeled vehicle in front of the black car.

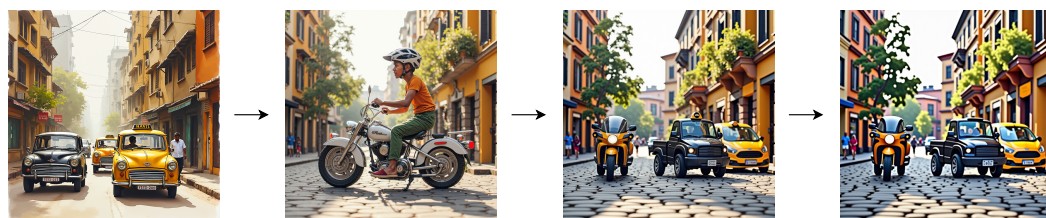

Figure 6: A failure trial with the same prompt. In the first and second rounds, although the actor provides seemingly correct edit prompts based on feedback, the Image Editor fails to migrate old elements to the new prompts. In the third round, the Image Editor fails to make any changes, resulting in a vague image that it is unable to properly interpret or refine. From our observations, most failure cases are caused by the Image Editor's inability to handle the vague outputs it produces itself.

## C CASE STUDY

We provide three groups of comparison of the image generated by these models in Figure 1. It can be observed that RL-RIG performs a stronger ability of following the extremely complex spatial descriptions in these examples, accurately capturing relationships that other models struggle with.

To better illustrate our framework's reasoning process, we present a detailed examination of the intermediate steps in two RL-RIG generation cases using the same prompt as described in Figure 5.

In the successful case in Figure 5, we can observe the complete Generate-Reflect-Edit cycle in action. The initial image generated by the Image Generator captures several spatial relationships correctly, but crucially misses two key requirements: the two-wheeled vehicle in front of the black car, and a person riding it. Next, the VLM Checker meticulously analyzes this initial output through a structured reasoning process, and identifies the missing relationships. The VLM Actor then formulates a targeted edit prompt that preserves correct elements while specifically requesting the missing components. Finally, the Image Editor successfully incorporates these elements, resulting in an image with all spatial requirements.

In contrast, under identical conditions but with a different random seed, Figure 6 illustrates a failed attempt where the editor either completely overwrites the original image, resulting in a fundamentally different scene, or makes no changes at all. This comparison highlights that our RL-RIG design is capable of reasoning and correcting unsatisfactory outputs from the base model, whereas its performance remains partially constrained by the limitations of the base model itself.

## D SCENE GRAPHS EXTRACTION

This part is realized by the LAION-SG dataset (Li et al., 2024). Specifically, for each ground-truth and generated image, GPT-4o is used to output structured JSON containing scene graph triplets in ["subject", "predicate", "object"] format, along with object and predicate lists, using the following prompt:

```
    Please extract the scene graph of the given image.
The scene graph just needs to include the relations of the salient
    objects and exclude the background.
The scene graph should be a list of triplets like ["subject", "
    predicate", "object"].
Both subject and object should be selected from the following list
    : {unique_items_list}.
The predicate should be selected from the following list: {
    unique_relations_list}.
Besides the scene graph, please also output the objects list in
    the image like ["object1", "object2", ..., "object"].
The object should be also selected from the above-mentioned object
     list. The output should only contain the scene graph and the
    object list.
```

```
Return the results in the following JSON format:
{{
    "scene_graph": [
        ["subject", "predicate", "object"],
        ...
    ],
    "object_list": [
        "object1", "object2", ...
    ],
    "predicate_list":[
        "predicate1", "predicate2", ...
    ]
}}
```

For evaluation, `unique_items_list` and `unique_relations_list` are limited only to the items that exist in the ground truth image's scene graph. Scene graph IoU, object IoU and relationship IoUs are then calculated using the common approach.

## E PROMPT OF VLM CHECKER

Here we provide the prompt of our VLM Checker.

```
How many of the following {length} descriptions does this image
    satisfy?
-----
{new_prompt}
-----
Please analyze step by step, and finally give the number of
    satisfied descriptions, ranging from 0 to {length}, in \\
    boxed{{}} format.
```

Regarding generalization, VLM Checker does have a general judgement ability, for it is substantially a generic VLM reasoning model without post training. As demonstrated in (Deng et al., 2025a), the base VLM model performs well on multiple tasks including image QA, detection, classification and maths, which proves its generalization capabilities.

## F MORE ABLATION STUDIES

To show that naive image editing approach does not work well, we add the following ablation studies.

1. Not using the Actor, feed original prompt to Image Editor iteratively.
2. Similar to 1 but only using unsatisfied constraints in the original prompt.
3. Generate 10 images and pick the best one by VLM Checker.

Table 3: Comparison of more ablation studies

| Method | SG-IoU | Ent-IoU | Rel-IoU | Qwen-Judge | GPT-Judge |
| --- | --- | --- | --- | --- | --- |
| Unsatisfied prompt | 0.3051 | 0.8176 | 0.7150 | 0.6739 | 0.5549 |
| Same prompt | 0.3357 | 0.9089 | 0.7632 | 0.7932 | 0.7234 |
| Pick 1 from 10 | 0.3468 | 0.9156 | 0.7729 | 0.8392 | 0.7273 |

It can be observed that prompt engineering approaches perform no better than the original Flux. We surmise that this phenomenon stems from two main factors. First, inversion-based image editing relies on guidance from the input text prompt. Repeatedly feeding the same prompt leads to identical

guidance signals, resulting in little to no improvement over single-pass editing. Moreover, each round of inversion and reversion introduces additional noise, which can degrade image quality. Second, the Image Editor is highly sensitive to the style of the edit prompt. In practice, directly reusing the original prompt often fails to produce the desired modifications, as it may not align well with the editor's preferred input style.

Although Pick 1 from 10 approach has a relatively fair performance, it is extremely computational expensive, whose inference time is almost a whole day. In fact, this result exactly fit our assumption that Reinforcement Learning is actually *pruning* unnecessary generation trajectories, instead of creating new ones. As a result, RL can transfer the performance of `pass@k` to `pass@1`, but can hardly optimize `pass@1` over `pass@k`. Additionally, our RL-RIG's generator can actually be based on 'Pick 1 from 10' if needed and further improve its performance.

## DECLARATION OF LLM USAGE

LLM was used only for polishing writing.

