# OpenReview forum: "Generative Spatial Reasoners via Reinforcement Learning based Intrinsic Reflection"
_ICLR.cc/2026/Conference — ICLR 2026 Conference Withdrawn Submission_

### Official Review · Reviewer_P7Y2 · 2025-10-24

**Soundness:** 1
**Presentation:** 1
**Contribution:** 1
**Rating:** 2
**Confidence:** 4

**Summary:**

The paper addresses the problem of spatial reasoning in text-to-image generation, observing that existing models often fail to capture complex spatial relationships despite producing high-quality images. To tackle this, the authors propose RL-RIG, a reinforcement learning framework that iteratively generate-reflect-edits image outputs. In each cycle, an initial image is generated from the text prompt, a vision-language model (VLM) checker evaluates whether all spatial constraints are met, and if not, an actor generates an edit prompt. An image editor then applies the edit, and the process repeats. On the LAION-SG dataset, the authors claim RL-RIG achieves up to 11% improvement in scene-graph IoU over strong open-source baselines

**Strengths:**

1. The idea of using an LLM-based checker and actor in an iterative loop is conceptually interesting.
2. Casting the LLM to propose edits based on where the image fails to meet the prompt could, in principle, improve compliance with complex prompts.

**Weaknesses:**

1. It is never clearly explained why reinforcement learning is the right tool. The paper simply defines an MDP with states (current image + target graph) and actions (prompts) but omits essential details (no discount factor, no criticism of episodic vs continuous tasks).

2. The only reward signal is the fraction of satisfied prompt requirements from the checker (Equation 2). This sparse, binary-like reward provides little guidance, yet no shaping or intermediate rewards are discussed. The group-relative scheme (GRPO) is mentioned, but the paper provides no intuition for its use, nor how rewards are actually computed or normalized across samples.

3. The experiments compare RL-RIG only to standalone generation models (Stable Diffusion, Flux) and a model trained on LAION-SG. There are no ablations or baselines that isolate components (e.g., a prompt-chain approach using the VLM without RL, or applying simple repeated editing without RL). For example, the “raw” RL-RIG (no post-training) is only compared via its final scores. It is unclear how much of the gain comes from the iterative framework itself versus the post-training fine-tuning.

4. No qualitative examples (except Figures 1 and 5), user studies, or diversity/quality metrics (like FID or user satisfaction) are given. The claim of “controllable spatial reasoning” is only supported by small numerical IoU gains (e.g., SG-IoU from 0.3319 to 0.3699 over Flux). This absolute improvement (~0.038) is modest, and it’s unclear if it is statistically significant.

5. The scene-graph IoU relies on automatically extracting objects/relations from images via GPT-4 (or similar), which may be error-prone. The “VLM-judge” metric uses the same Qwen2.5 model (ReFT) that is involved in the system, potentially biasing scores. In essence, the method is evaluated by the models it uses internally, raising concerns of circularity.

6. For example, the notion of “intrinsic reflection” (and even “chain-of-thought” for images) is introduced in the contribution list, but not clearly defined. Some wording is confusing or erroneous (“As shown in shown in Figure 2”).

7. As it is an iterative process, the runtime of the process will be very high and not reported in the paper.

**Questions:**

1. What is the novelty of the paper? Just training VLMs on LAION-SG and use of VLMs repeatedly in a circular manner.
2. Equations (11–14) merely repeat standard PPO-like updates. How does group sampling help here?
3. The actor model (a fine-tuned LLM) is generating text prompts; How could this be treated as an RL policy?
4. The “raw” RL-RIG (no training) is only tested against the post-trained version, but the gain is small (+3.5% SG-IoU over Flux). It is unclear what this ablation tells us

**Details Of Ethics Concerns:**

There are no explicit ethical statements in the paper, as there are not many qualitative figures. It is not applicable for this paper.

---

### Official Review · Reviewer_UHvN · 2025-10-28

**Soundness:** 2
**Presentation:** 2
**Contribution:** 2
**Rating:** 2
**Confidence:** 5

**Summary:**

The paper proposes a generate-reflect-edit paradigm to improve the instruction following ability for spatial relationships in complex prompts in text-to-image generation. Experiments are conducted on a subset of LAION-SG dataset and quantitatively compared with three other models.

**Strengths:**

The proposed approach systematically tackles the lack of adherence to spatial relationships in generated images and shows notable improvements over three recent methods.

**Weaknesses:**

The proposed method is not new and simply applies R1 style training for the task of adhering to spatial constraints in image generation without providing substantial new insights. It relies on the computationally expensive and complex RL pipeline, which relies on sparse reward signals to enhance adherence to spatial relationships described in text prompts.

The major concern is evaluation. Missing quantitative and qualitative comparisons with prior works that use scene graphs or bounding boxes or other layouts [1-6]  for conditioning the image generation (with layouts generated from LLMs from text prompts).

The proposed method is model-specific and requires retraining for every new model.

The improvements shown in Table 1 over the raw version (without training) is marginal and so the effectiveness of the proposed method is limited compared to the training costs involved.

Since the raw version of the proposed method performs well, it can also be achieved by efficient sampling from the diffusion model (e.g., best one out of N). So, it needs to be compared with other efficient sampling methods to show the effectiveness of the proposed method.

Computational cost during inference is not reported. Since it is an iterative process, it is expensive and so the trade-off of accuracy and computation should be discussed.

Missing ablation studies on the number of responses considered during GRPO training. Does scaling it improve the adherence and quality further?

Although the method improves the IoU metric, its influence on the image quality is not known. So, additional evaluations for the standard image quality metrics (FID, CLIP-score) should be reported as well.

Qualitative results are limited. Only three examples are shown. More qualitative results should be included in the paper.

Currently, the paper is incomplete with insufficient experiments, missing ablation studies and insufficient evaluation. So, it needs significant revisions before it can be considered for acceptance although the idea is promising.

[1] Xu et. al. Joint Generative Modeling of Grounded Scene Graphs and Images via Diffusion Models, TMLR 2025

[2] Farshad et. al. SceneGenie: Scene Graph Guided Diffusion Models for Image Synthesis

[3] Li et. al. Gligen: Open-set grounded text-to-image generation, CVPR 2023

[4] Qu et. al. LayoutLLM-T2I: Eliciting Layout Guidance from LLM for Text-to-Image Generation

[5] Zhao et. al. Uni-controlnet: All-in-one control to text-to-image diffusion models, NeurIPS 2023

[6] Cheng et. al. HiCo: Hierarchical Controllable Diffusion Model for Layout-to-image Generation, NeurIPS 2024

**Questions:**

See weaknesses above

Since the image generator is not updated during training, does the method work when swapped with other models at inference?

How about training only the modules in a single phase? Is two phase essential?

---

### Official Review · Reviewer_z5Es · 2025-10-28

**Soundness:** 2
**Presentation:** 2
**Contribution:** 2
**Rating:** 4
**Confidence:** 2

**Summary:**

This paper introduces a Generate–Reflect–Edit framework that enhances spatial reasoning in text-to-image generation. It integrates reinforcement learning and vision–language models (VLMs) to iteratively refine generated images based on spatial consistency feedback, using Group Relative Policy Optimization (GRPO) for training. Experiments on the LAION-SG dataset demonstrate that RL-RIG achieves up to 11% improvement in spatial accuracy over state-of-the-art models like Flux and Stable Diffusion 3.5.

**Strengths:**

(1) Proposes a novel Generate–Reflect–Edit framework that combines reinforcement learning with intrinsic reflection to address the “spatial reasoning dilemma” in image generation.

(2) Provides thorough experimental validation with new spatial reasoning metrics (Scene Graph IoU, VLM-as-a-Judge), demonstrating consistent and interpretable gains over strong baselines.

**Weaknesses:**

(1) The approach is heavily dependent on the reasoning and editing capacity of underlying base models (e.g., Flux, VLMs), limiting generalization.

(2) The training setup is computationally demanding, and results rely on a specific dataset (LAION-SG) that may not generalize to broader visual domains.

(3) The paper could better clarify the reflection mechanism’s scalability and failure cases, particularly when the Image Editor fails to make meaningful edits.

**Questions:**

None

---

### Official Review · Reviewer_o7aW · 2025-11-01

**Soundness:** 3
**Presentation:** 3
**Contribution:** 3
**Rating:** 8
**Confidence:** 3

**Summary:**

This paper presents RL-RIG, a framework designed to tackle the spatial reasoning dilemma in text-to-image generation, where existing models often fail to accurately represent complex spatial relationships described in prompts. RL-RIG introduces a "Generate-Reflect-Edit" paradigm, leveraging reinforcement learning and Vision-Language Models (VLMs) to iteratively refine images until all spatial constraints are met.
The system comprises four key components: an Image Generator, a VLM Checker, a VLM Actor, and an Image Editor. Training is conducted in two phases, optimizing the VLM Actor and Image Editor using Group Relative Policy Optimization (GRPO). The components are defined after a detailed analysis of the existing drawbacks in the classical pipelines for image generation.
Empirical results on the LAION-SG dataset show that RL-RIG significantly outperforms state-of-the-art models such as Stable Diffusion 3.5 Large and Flux, achieving up to 11% improvement in spatial reasoning metrics like Scene Graph IoU and VLM-based evaluations.

**Strengths:**

It is an interesting work.

A solid analysis of the problem is provided, identifying the challenges in image generation regarding the robustness in front of complex spatial relations in the prompt.

The work tackles a critical challenge in image generation: fine-grained spatial reasoning. While many models focus on visual fidelity, RL-RIG emphasizes structural integrity and semantic alignment, which is a meaningful challenge.

RL-RIG employs Scene Graph IoU and a VLM-as-a-Judge strategy, which are more aligned with spatial reasoning than conventional pixel-level metrics.

11% improvement over state-of-the-art open-source models on the LAION-SG dataset is reported.

**Weaknesses:**

In general, the paper is quite solid, I don’t appreciate major weaknesses.

The evaluation conducted on the LAION-SG dataset, may not capture the full diversity of spatial reasoning challenges (e.g., indoor scenes, object occlusion, hierarchy in relations). Exploration and broader testing on other datasets would strengthen claims of generalizability.

There are some parts of the paper that are difficult to follow (for example the section on training process), and it may hinder reproducibility. Clarifying these aspects would enhance the paper’s accessibility and practical impact.

**Questions:**

The use of Scene Graph IoU is simple and effective. However I am wondering if it is really a structural measure. Two images might have identical scene graphs (same objects and relations), but the actual spatial arrangement (e.g., left vs. right, near vs. far) could differ significantly. SG-IoU may yield a high score even if object positions are swapped, as long as the relationships are preserved. Have you studied other graph-based metrics like Weisfeiler Leman graph isomorphism test, or Visual Genome based metrics?

What is the limit of the proposed approach regarding the complexity of relations? Is it able to capture complex and hierarchical relations? (For example, let me suggest the following prompt “a man and a woman are riding a motorcycle, that is running in a parallel line of a road next to a red car crossing a bridge. The bridge is over a river, and a boat is passing under it”. In terms of a scene graph, it contains different objects connected by different types of relations.

---

### Note · Authors · 2025-11-12

**Comment:**

Dear all,

After careful consideration, we have decided to withdraw this paper from ICLR2026. The main reason for this request is that the current set of reviews shows substantial divergence in evaluations. We express gratitude to all the reviewers' time and feedback, and will further refine this paper based on these suggestions.

Thanks for your understanding.

Best regards,

All Authors.

**Withdrawal Confirmation:**

I have read and agree with the venue's withdrawal policy on behalf of myself and my co-authors.